# Association between serum 25-hydroxyvitamin D levels and Early Vascular Aging in young and middle-aged adults

**Jinpeng Cong**[1], **Rui Hu**[2]*, **Jinyan Ren**[2], **Xinfeng Wang**[2], **Na Li**[2], **Ying Sun**[2]

1 Department of Respiratory and Critical Care Medicine, Affiliated Hospital of Qingdao University, Qingdao, China, 2 Department of Health Management Center, Affiliated Hospital of Qingdao University, Qingdao, China

* doctorhurui@163.com

## Abstract

### Background

Cardiovascular disease (CVD) remains the leading cause of mortality and morbidity worldwide. Early vascular aging (EVA) is an independent predictor of cardiovascular risk in young and middle-aged adults. The plausible association between EVA and vitamin D warrants further investigation.

### Methods

This study examined the relationship between serum 25-hydroxyvitamin D [25(OH)D] levels and EVA in young and middle-aged healthy adults. This cross-sectional study included 2047 eligible participants who underwent physical examinations at the Health Management Center of the Affiliated Hospital of Qingdao University between May 2023 and May 2025. Participants were categorized into EVA group (n = 687) and control group (n = 1360) based on brachial ankle pulse wave velocity (baPWV). Logistic regression and restricted cubic splines assessed the association between 25(OH)D and EVA, with an inflection point identified using two-piecewise linear regression. Subgroup analyses and interaction tests were conducted by age, sex, blood collection month, smoking status, alcohol drinking status, hypertension, diabetes, and body mass index (BMI).

### Results

The prevalence of EVA was 33.56%. Each 10 ng/mL increase in 25(OH)D was associated with a 19% decrease in the likelihood of EVA (OR = 0.81, 95% CI: 0.70–0.94, P = 0.008). Compared with Q1, Q4 had a significantly lower risk (OR = 0.55, 95% CI: 0.40–0.76, P < 0.001), while Q2 and Q3 did not differ (both P > 0.05). Restricted cubic spline analysis showed a nonlinear L-shaped association (P for non linearity = 0.015) with a threshold at 17.90 ng/mL: below this level, higher 25(OH)D was linked to a

**Data availability statement:** All relevant data are within the manuscript and its Supporting Information files S1 Dataset.

**Funding:** The author(s) received no specific funding for this work.

**Competing interests:** The authors have declared that no competing interests exist.

lower EVA risk (OR=0.90, 95%CI: 0.85–0.96), whereas no significant association was observed above it. Subgroup analyses indicated a particularly significant inverse association between 25(OH)D levels and EVA in men and overweight/obese individuals. Significant interactions were observed for sex, BMI, and age (all P<0.05), while interactions with other factors, such as blood collection month, hypertension, diabetes, smoking status, drinking status were not statistically significant.

## Conclusion

This study suggests an inverse association between 25(OH)D levels and EVA, with a more pronounced association possibly observed in men and individuals with overweight/obesity, and further indicates a threshold at 17.9 ng/mL below which this inverse association is significantly enhanced.

## Introduction

Cardiovascular disease (CVD) remains the predominant global cause of mortality and morbidity. Epidemiological projections suggest that annual CVD-related mortality will exceed 35 million by 2050, posing an unprecedented challenge to global public health strategies [1]. Vascular aging serves as a pivotal pathological foundation for the onset and progression of CVD, as well as facilitating the development of prevalent conditions like hypertension, diabetes, and chronic kidney disease. Hence, prioritizing attention to vascular aging is crucial for the prevention and management of CVD. Early vascular aging (EVA) denotes the premature manifestation of vascular aging, characterized by a more accelerated process compared to normal aging, leading to a discernible gap between biological and chronological vascular age [2]. Distinguished from typical atherosclerosis, EVA emphasizes accelerated vascular aging processes at earlier stages and their associated risk phenotypes, serving as an independent predictor of cardiovascular risk [3–5].Pulse wave velocity (PWV) is a central measure of vascular aging, with brachial–ankle PWV (baPWV) commonly used in clinical and epidemiological settings [6,7]. Vitamin D, a fat-soluble vitamin, is predominantly stored as 25-hydroxyvitamin D [25(OH)D]. Research indicates that vitamin D is involved not only in regulating calcium and phosphorus metabolism but also in protecting arterial endothelial function, and is associated with a lower risk of atherosclerosis and CVD [8–10]. Although previous studies have examined the association between vitamin D and atherosclerosis [11–13], they predominantly employed fixed cut-off values without accounting for age- and sex-specific stratification, thereby limiting their ability to identify premature vascular aging. While EVA addresses this limitation through individualized definitions, evidence regarding its relationship with vitamin D remains limited. Moreover, while vitamin D deficiency has been linked to higher cardiovascular risk, recent large-scale trials have not conclusively demonstrated cardiovascular benefits of vitamin D supplementation [14–17].

This discrepancy between observational associations and interventional outcomes suggests that the relationship between vitamin D and vascular health may involve threshold effects or population heterogeneity.

Therefore, further exploration of the association between vitamin D and EVA holds significant scientific importance and practical relevance. This study aimed to explore the association between serum 25(OH)D levels and EVA in young and middle-aged adults undergoing physical examinations, to provide new insights into the occurrence of vascular aging in this population.

## Methods

### Study population

This cross-sectional study initially screened 9,816 adults who underwent baPWV measurement at the Health Management Center of the Affiliated Hospital of Qingdao University between May 2023 and May 2025. Eligible participants were aged ≥18 and <60 years and had complete data for 25(OH)D, baPWV, and other routine biochemical covariates. Individuals were excluded if they had: 1) severe medical disorders, such as infectious diseases, coronary atherosclerotic heart disease, cerebral infarction, malignant tumors, and liver or kidney insufficiency; 2) conditions affecting serum 25(OH)D levels, such as parathyroid disease, autoimmune kidney diseases, severe liver diseases, gastrointestinal disorders, history of gastrointestinal surgery, rickets, and burns; 3) pregnancy or lactation; 4) recent use of calcium agents, vitamin D supplements, or medications influencing serum 25(OH)D synthesis or metabolism (such as phenytoin sodium, phenobarbital, isoniazid, glucocorticoids) within the past 3 months. Ultimately, 2,047 participants (687 in the EVA group and 1,360 in the control group) were included in the final analysis. Data collection took place from September 1 to September 5, 2025. The inclusion process is illustrated in S1 Fig in S2 File.

This study was approved by the Ethics Committee of the Affiliated Hospital of Qingdao University (approval No. QYFY WZLL30489) on August 28, 2025. The research was conducted in accordance with the principles of the Declaration of Helsinki, local laws, and institutional requirements. Due to its retrospective nature and the use of de-identified data extracted from existing medical records, the requirement for informed consent was waived by the ethics committee.

### Assessment of serum 25(OH)D levels

Venous blood samples were collected in the morning and serum was isolated through centrifugation. Serum 25(OH)D levels were measured using a delayed one-step immunoassay with chemiluminescence assay (CMIA) technology on an ARCHITECT automatic chemiluminescence analyzer (i200, Abbott Diagnostics, USA). The intra- and inter-assay coefficients of variation for this method were <5%. Regular calibration was performed using 25-OH Vitamin D controls to ensure comparability and accuracy. Based on serum 25(OH)D levels, participants were stratified into four quartile groups: Q1 (<13.6 ng/mL), Q2 (13.7–18.1 ng/mL), Q3 (18.2–23.6 ng/mL), and Q4 (≥23.7 ng/mL).

### Assessment of EVA using baPWV

The blood pressure pulse wave was assessed using an automatic waveform analyzer (BP-203RPE III, Omron Healthcare, Japan). After participants rested supine for 10 minutes, an oscillographic blood pressure cuff was applied to their limbs, and bilateral baPWV values were averaged. The baPWV reference curves were derived from the standard database built into the Omron BP-203RPE III device. The baPWV reference curves were derived from the built-in standard database of the Omron BP-203RPE III device. This database was established from a large health-check population of Chinese healthy adults and was stratified and adjusted by age and sex to provide reference values (Mean, + 1SD, and +2SD) (S2 Fig in S2 File). Based on existing literature [18–21], EVA was defined as a baPWV exceeding two standard deviations above the mean for the same sex and age group. The study participants were divided into the EVA group (n = 687) and the control group (n = 1360) based on this criterion.

## Assessment of covariates

Demographic information was collected from all participants, including age, sex, blood collection month, height, and weight. Body mass index (BMI) was calculated, and blood pressure measurements including systolic blood pressure (SBP) and diastolic blood pressure (DBP) were recorded. Medical history data were obtained through standardized questionnaires, including information on underlying diseases, smoking and drinking status habits. Laboratory tests were conducted after overnight fasting (≥8 hours) and included measurements of triglycerides (TG, mmol/L), total cholesterol (TC, mmol/L), low-density lipoprotein cholesterol (LDL-C, mmol/L), high-density lipoprotein cholesterol (HDL-C, mmol/L), fasting blood glucose (FBG, mmol/L), glycated hemoglobin (HbA1c, %), uric acid (UA, µmol/L), creatinine (Cr, µmol/L), and homocysteine (Hcy, µmol/L). Hypertension was defined as SBP ≥ 140 mmHg and/or DBP ≥ 90 mmHg, or prior diagnosis with antihypertensive treatment [22]. Diabetes was defined as FBG ≥ 7 mmol/L and/or HbA1c ≥ 6.5%, or existing treatment [23]. Smoking and alcohol drinking statushistories were defined as ≥1 unit/day for more than 6 consecutive months, or having previously met criteria but quit for less than 6 months. According to World Health Organization, BMI was categorized as underweight (<18.5 kg/m²), normal weight (18.5–24.9 kg/m²), overweight (25.0–29.9 kg/m²), and obese (≥30.0 kg/m²) [24].

## Statistical analysis

Statistical analyses were performed using R statistical software (version 4.4.1). Data normality was assessed using quantile-quantile (Q-Q) plots. Normally distributed continuous variables were expressed as mean ± standard deviation and compared using independent Student's t-tests. Non-normally distributed continuous variables were presented as median (25th and 75th percentiles) and analyzed using Mann-Whitney U tests. Categorical variables were reported as frequencies and percentages, and compared using Pearson's chi-square test. Subsequently, three multivariate logistic regression models were developed to assess the association between 25(OH)D and EVA: Model 1 was unadjusted; Model 2 was adjusted for age, sex, blood collection month, BMI, smoking, alcohol drinking, hypertension, and diabetes. Combining variance inflation factors (VIF) to evaluate multicollinearity among independent variables, Model 3 further adjusted for SBP, DBP, TG, HDL-C, FBG, UA, Cr, HbA1c, and Hcy on the basis of Model 2. Restricted cubic spline (RCS) curve analysis was further performed to evaluate the dose-response relationship between 25(OH)D and EVA. The 5th, 35th, 65th, and 95th percentiles of 25(OH)D levels were selected as knots, with the median serving as the reference value. In cases where a non-linear association was indicated by the RCS analysis, a two-piecewise segmented regression model was fitted using the segmented package to estimate the potential threshold effect. Specifically, the threshold was estimated from the data under a one-breakpoint model (npsi = 1), with the initial value of the breakpoint parameter (psi) set to the median of 25(OH)D. The optimal threshold was selected as the value that minimized the Akaike Information Criterion (AIC) and Bayesian Information Criterion (BIC). In addition to AIC/BIC, the standard errors of model parameters were considered to reflect estimation precision. The resulting threshold was 17.9 ng/mL.Finally, subgroup analysis, forest plots, and interaction analysis were performed to assess the association between 25(OH)D and EVA across different subgroups. All statistical tests were two-sided, with P < 0.05 considered statistically significant.

## Results

### Baseline characteristics of the participants

Table 1 displays data from a study encompassing 2047 participants, with an average age of 47.58 ± 8.15 years, comprising 1228 males (59.99%) and 819 females (40.01%). The EVA group consisted of 687 cases, while the control group comprised 1360 cases, resulting in a prevalence rate of 33.56%.Compared to the control group, the EVA group exhibited a higher proportion of males and older average age. Additionally, the EVA group demonstrated a higher prevalence of hypertension, diabetes, smoking, and alcohol drinking, as well as higher levels of BMI, SBP, DBP, TG, TC, LDL-C, FBG,

**Table 1. Participant characteristics based on EVA status.**

| Characteristic | Total (n = 2047) | Control group (n = 1360) | EVA Group(n = 687) | t/Z/χ² | p-value |
|---|---|---|---|---|---|
| Age (years) | 47.58 ± 8.15 | 46.48 ± 8.26 | 49.76 ± 7.48 | 9.052 | **<0.001** |
| Sex, n(%) | | | | 17.565 | **<0.001** |
| Male | 1228 (59.99%) | 772 (56.77%) | 456 (66.38%) | | |
| Female | 819 (40.01%) | 588 (43.23%) | 231(33.62%) | | |
| BMI(kg/m²) | 25.48 ± 3.64 | 25.07 ± 3.52 | 26.30 ± 3.75 | −7.253 | **<0.001** |
| Hypertension, n(%) | 521(25.45%) | 198(14.56%) | 323(47.02%) | 253.421 | **<0.001** |
| Diabetes, n(%) | 206(10.06%) | 91(6.70%) | 115(16.74%) | 50.834 | **<0.001** |
| Smoking status, n (%) | 395 (19.30%) | 234 (17.21%) | 161 (23.44%) | 11.373 | **<0.001** |
| Alcohol drinking status, n(%) | 373 (18.22%) | 207 (15.22%) | 166 (24.16%) | 24.494 | **<0.001** |
| Blood collection month, n(%) | | | | 0.487 | 0.485 |
| May to October | 1043(50.95%) | 685(50.37%) | 358(52.11%) | | |
| November to April | 1004(49.05%) | 675(49.63%) | 329(47.89%) | | |
| 25(OH)D(ng/mL) | 19.38 ± 8.06 | 19.62 ± 7.93 | 18.90 ± 8.28 | −2.865 | 0.043 |
| 25(OH)D Quartile Groups, n(%) | | | | 9.070 | 0.028 |
| Q1 | 523(25.55%) | 338(24.85%) | 185(26.93%) | | |
| Q2 | 504(24.62%) | 329(24.19%) | 175(25.47%) | | |
| Q3 | 515(25.16%) | 330(24.27%) | 185(26.93%) | | |
| Q4 | 505(24.67%) | 363(26.69%) | 142(20.67%) | | |
| SBP (mmHg) | 124.14 ± 15.78 | 119.34 ± 13.17 | 133.63 ± 16.21 | 20.019 | **<0.001** |
| DBP(mmHg) | 76.97 ± 11.71 | 73.54 ± 10.16 | 83.77 ± 11.62 | 19.617 | **<0.001** |
| TG (mmol/L) | 1.23(0.84, 1.86) | 1.13(0.76, 1.69) | 1.49(1.02, 2.24) | 10.411 | **<0.001** |
| TC (mmol/L) | 5.37 ± 1.06 | 5.33 ± 1.05 | 5.46 ± 1.06 | 2.608 | 0.009 |
| LDL-C (mmol/L) | 3.09 ± 0.83 | 3.06 ± 0.82 | 3.14 ± 0.84 | 2.245 | 0.025 |
| HDL-C (mmol/L) | 1.62 ± 0.37 | 1.66 ± 0.38 | 1.55 ± 0.34 | −6.308 | **<0.001** |
| FBG (mmol/L) | 5.09(4.74, 5.57) | 5.01(4.67, 5.44) | 5.29(4.89, 5.88) | 10.041 | **<0.001** |
| HbA1c(%) | 5.70(5.50, 5.80) | 5.70(5.40, 5.80) | 5.75(5.50, 6.00) | 7.440 | **<0.001** |
| Cr (μmol/L) | 88.82 ± 20.97 | 87.87 ± 17.08 | 90.70 ± 26.99 | 2.509 | 0.012 |
| UA (μmol/L) | 368.89 ± 93.39 | 361.20 ± 91.29 | 384.13 ± 95.67 | 5.200 | **<0.001** |
| Hcy (μmol/L) | 10.60(9.03,12.16) | 10.32(8.81,11.89) | 10.97(9.44,12.69) | 5.811 | **<0.001** |
| ba-PWV(cm/s) | 1362.97 ± 208.74 | 1252.15 ± 121.96 | 1582.35 ± 167.18 | 45.958 | **<0.001** |

Data are expressed as mean ± standard deviation, median (interquartile range), and number of cases (%). EVA, early vascular aging; BMI, body mass index; SBP, systolic blood pressure; DBP, diastolic blood pressure; TG, triglyceride; TC, total cholesterol; LDL-C, low-density lipoprotein cholesterol; HDL-C, high-density lipoprotein cholesterol; FBG, fasting blood glucose; HbA1c, glycated hemoglobin; Cr, Creatinine; UA, uric acid; Hcy, Homocysteine; P < 0.05 are bolded.

HbA1c, Cr, UA, and Hcy compared to the control group (all P < 0.05), while serum 25(OH)D levels and HDL-C were lower in the EVA group (all P < 0.05). There was no statistically significant difference in blood collection months between the two groups. Quartile analysis of serum 25(OH)D concentrations revealed that the EVA positive rates in the lower quartiles (Q1: 26.93%, Q2: 25.47%, Q3: 26.93%) were significantly higher than the highest quartile (Q4: 20.67%), with statistically significant differences observed across these subgroups (P < 0.05).

## Association analysis between serum 25(OH)D levels and EVA

In this study, VIF were employed to assess the degree of multicollinearity in the multiple linear regression model. TC and LDL-C demonstrated high multicollinearity (VIF > 10), thus they were excluded from Model 3. The remaining covariates

had VIF values of less than 5, indicating that there was no significant multicollinearity among the regression model (S1 Table).Table 2 displays the outcomes of a multivariate logistic regression analysis examining the relationship between 25(OH) D levels and EVA incidence. The findings revealed that in the fully adjusted model for continuous 25(OH)D, each 10 ng/mL increase in 25(OH)D was associated with a 19% decrease in the likelihood of EVA (OR = 0.81, 95% CI: 0.70–0.94, P = 0.008). Furthermore, to validate the reliability of the outcomes, we conducted a sensitivity analysis by categorizing 25(OH)D into quartiles (Q1, Q2, Q3, and Q4) and analyzing these across the three regression models (Model 1, Model 2, and Model 3). Compared to the lowest quartile (Q1), the Q4 group consistently showed significantly reduced EVA risk across all three models, with ORs of 0.71 (95% CI: 0.55–0.93, P = 0.013) in Model 1, 0.55 (95% CI: 0.40–0.75, P < 0.001) in Model 2, and 0.58 (95% CI: 0.42–0.80, P < 0.001) in Model 3, respectively. In contrast, no statistically significant differences in EVA were observed between Q2 and Q1, or between Q3 and Q1, across all three models (all P > 0.05).

## Non-linear relationship and optimal threshold of 25(OH)D for EVA

The association between serum 25(OH)D levels and EVAwas initially evaluated through RCS curve analysis within a multivariate model (Model 3), adequately adjusted for covariates. The results revealed a statistically significant non-linear relationship between 25(OH)D levels and EVA risk, demonstrating an L-shaped pattern (P-non-linear = 0.015, Fig 1). Further analysis using a two-stage linear regression model identified a serum 25(OH)D threshold of 17.90 ng/mL. Below this threshold, increased 25(OH)D levels were significantly associated with a reduction in EVA(OR=0.90, 95%CI: 0.85–0.96). Conversely, above the threshold, the risk reduction trend plateaued, and the association became statistically nonsignificant (OR=0.98, 95%CI: 0.97–1.01).

## Subgroup analyses and interaction tests

To further investigate the association between serum 25(OH)D levels and EVA across different demographic groups, we conducted subgroup analyses based on age, sex, blood collection month, smoking and alcohol drinking status, hypertension, diabetes, and BMI using a fully adjusted multivariate logistic regression model (Model 3) to account for confounding variables. As depicted in Fig 2, a noteworthy inverse association between increased 25(OH)D levels and reduced EVAwas particularly evident among males (OR = 0.96, 95% CI: 0.94–0.98, P < 0.001) and individuals classified as overweight or obese (BMI ≥ 25, OR = 0.96, 95% CI: 0.94–0.98, P < 0.001). Additionally, similar trends of reduced EVA associated with 25(OH)D levels were observed in individuals aged 40–50 years, in those with blood collection month from May to October, and among populations without diabetes or hypertension, as well as in individuals with a history of smoking or alcohol

Table 2. Multivariate logistic regression analysis of the association between serum 25(OH)D and EVA.

| | Model 1 | | Model 2 | | Model 3 | |
|---|---|---|---|---|---|---|
| Characteristic | OR(95% CI) | P value | OR(95% CI) | P value | OR(95% CI) | P value |
| 25(OH)D(per 10 ng/mL) | 0.89(0.79, 1.00) | 0.059 | 0.79(0.68, 0.90) | 0.001 | 0.81 (0.70, 0.94) | 0.008 |
| Q1 | Reference | | Reference | | Reference | |
| Q2 | 0.97(0.75, 1.26) | 0.827 | 0.87(0.65, 1.16) | 0.346 | 0.84(0.62, 1.13) | 0.256 |
| Q3 | 1.02(0.79, 1.32) | 0.853 | 0.91(0.68, 1.22) | 0.529 | 0.96(0.71, 1.29) | 0.767 |
| Q4 | 0.71(0.55, 0.93) | **0.013** | 0.55(0.40, 0.75) | **<0.001** | 0.58(0.42, 0.80) | **<0.001** |

ORs for continuous 25(OH)D represent the change per 10 ng/mL increase. For quartile analysis, ORs compare each quartile to Q1 (reference). Model 1: unadjusted; Model 2: adjusted for age, sex, BMI, hypertension, diabetes, smoking status, alcohol drinking status and blood collection month; Model 3: adjusted for model 2 and SBP, DBP, TG, HDL-C, FBG, HbA1c, Cr, UA, Hcy. OR, odds ratio; CI, confidence interval. P<0.05 are bolded.

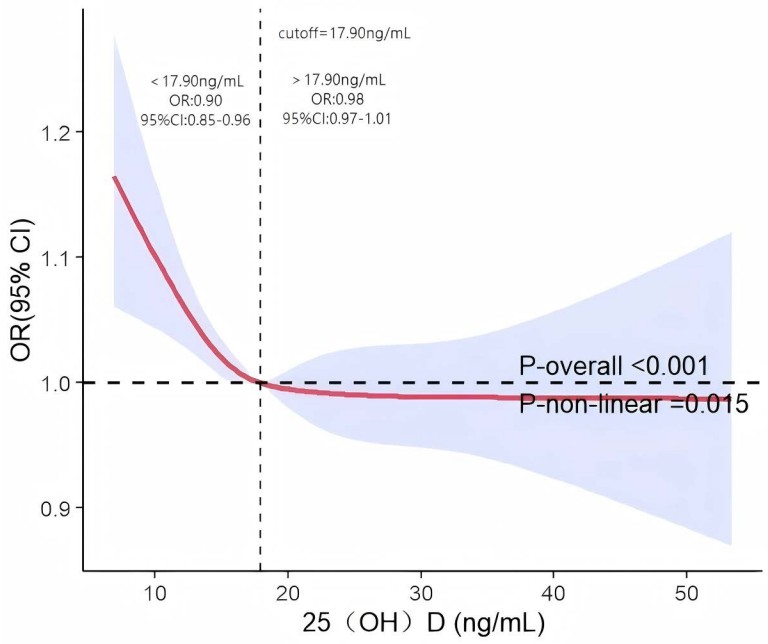

**Fig 1. Restricted cubic spline curve (RCS) plot of the relationship between serum 25(OH)D and EVA.** Odds ratios (ORs) represent the change per 1 ng/mL increase in serum 25(OH)D. The solid red line represents the OR with cutoff values indicated at 17.90 ng/mL. For serum 25(OH)D levels below 17.90 ng/mL, OR = 0.90 (95% CI: 0.85-0.96), which indicates a negative association with EVA risk. For levels above 17.90 ng/mL, OR = 0.98 (95% CI: 0.97-1.01), suggesting minimal impact on EVA risk. Adjustment factors are consistent with Model 3.

drinking status. Further interaction analysis revealed significant intergroup interactions for age, sex, and BMI (all P interaction values < 0.05). However, no statistically significant interactions were found for factors such as blood collection month, smoking, alcohol drinking status, hypertension, and diabetes.

## Discussion

Vascular aging is a multifactorial process driven by endothelial dysfunction, vascular smooth muscle cell (VSMC) aging, perivascular adipose tissue inflammation, and arterial calcification [25], and it is mainly characterized by decreased vascular compliance and sclerosis. Vascular aging that occurs at an earlier age, with greater severity and faster progression, is referred to as EVA. EVA begins in the embryonic period, and changes in vascular structure, increased stiffness, and endothelial dysfunction are observed at an earlier age. This pathological aging state is prone to lead to the occurrence of the chain of aging cardiovascular events such as premature cardiovascular disease(CVD), stroke, and peripheral artery disease [26]. Research indicates a 1.7-fold increase in the risk of cardiovascular events in individuals with EVA [3]. Niiranen et al. [27] found that even among hypertensive patients treated with antihypertensive medications, there remains a 50% residual risk of cardiovascular mortality. They proposed that EVA is one of the causes of this residual risk. Therefore,the concept of EVA has been introduced into the primary prevention of cardiovascular disease,where timely detection,diagnosis,and intervention are crucial to mitigate the onset and progression of vascular pathology. Currently, the assessment of vascular aging primarily relies on pulse wave velocity (PWV) and carotid intima-media thickness (cIMT). PWV is widely regarded as the premier metric for gauging vascular aging, surpassing traditional indicators like blood pressure, blood glucose, and lipid levels in predicting cardiovascular events. Brachial-ankle Pulse Wave Velocity (baPWV) is commonly utilized to assess the stiffness of large and medium arteries, with higher baPWV values indicating greater arterial stiffness. This study assessed baPWV in a cohort of middle-aged and young individuals, defining EVA as baPWV exceeding two

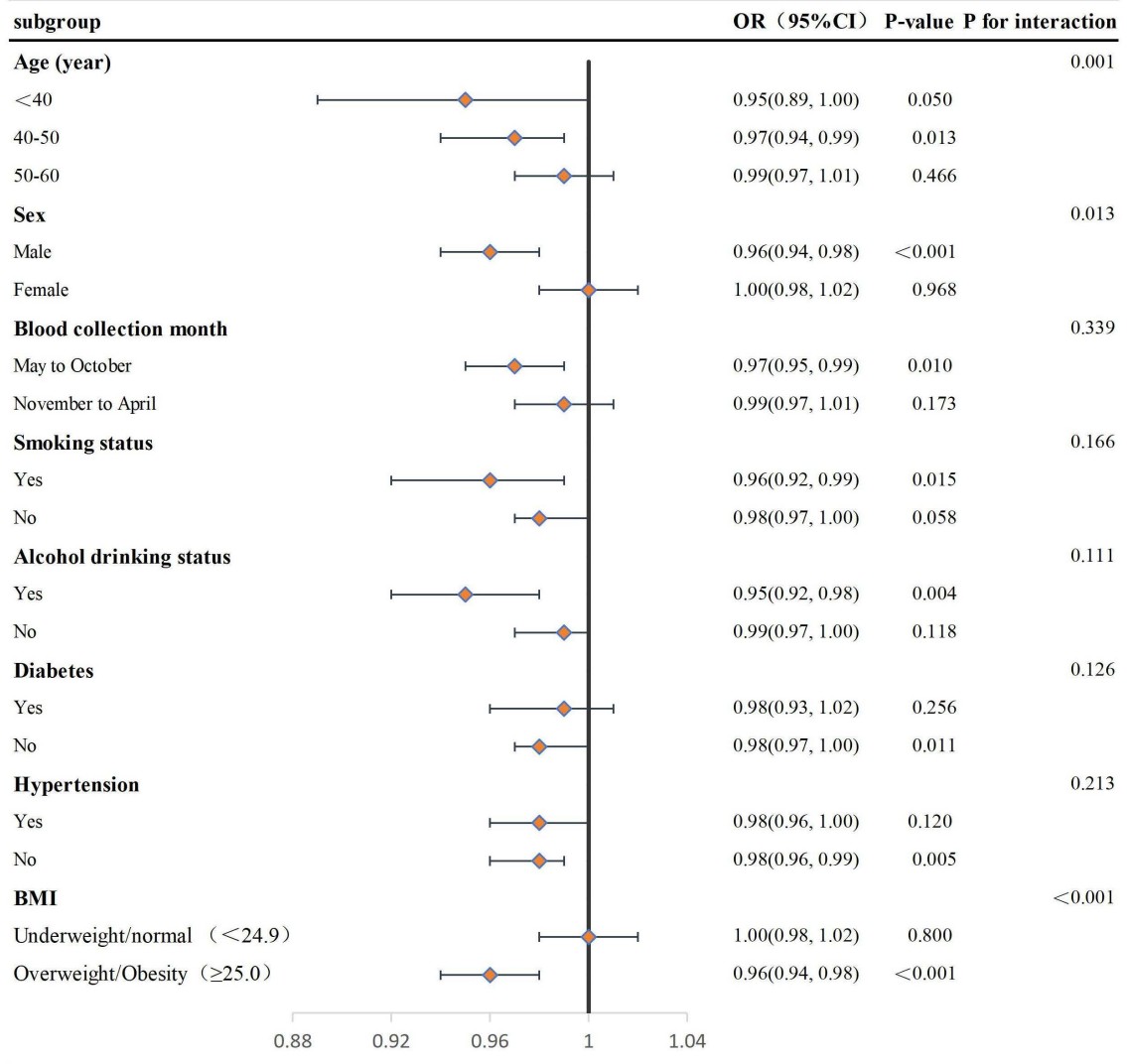

| subgroup | OR（95%CI） | P-value | P for interaction |
|---|---|---|---|
| **Age (year)** | | | 0.001 |
| <40 | 0.95(0.89, 1.00) | 0.050 | |
| 40-50 | 0.97(0.94, 0.99) | 0.013 | |
| 50-60 | 0.99(0.97, 1.01) | 0.466 | |
| **Sex** | | | 0.013 |
| Male | 0.96(0.94, 0.98) | <0.001 | |
| Female | 1.00(0.98, 1.02) | 0.968 | |
| **Blood collection month** | | | 0.339 |
| May to October | 0.97(0.95, 0.99) | 0.010 | |
| November to April | 0.99(0.97, 1.01) | 0.173 | |
| **Smoking status** | | | 0.166 |
| Yes | 0.96(0.92, 0.99) | 0.015 | |
| No | 0.98(0.97, 1.00) | 0.058 | |
| **Alcohol drinking status** | | | 0.111 |
| Yes | 0.95(0.92, 0.98) | 0.004 | |
| No | 0.99(0.97, 1.00) | 0.118 | |
| **Diabetes** | | | 0.126 |
| Yes | 0.98(0.93, 1.02) | 0.256 | |
| No | 0.98(0.97, 1.00) | 0.011 | |
| **Hypertension** | | | 0.213 |
| Yes | 0.98(0.96, 1.00) | 0.120 | |
| No | 0.98(0.96, 0.99) | 0.005 | |
| **BMI** | | | <0.001 |
| Underweight/normal（<24.9） | 1.00(0.98, 1.02) | 0.800 | |
| Overweight/Obesity（≥25.0） | 0.96(0.94, 0.98) | <0.001 | |

**Fig 2. Subgroup analysis of the relationship between serum 25(OH)D and EVA.** Odds ratios (ORs) represent the change per 1 ng/mL increase in serum 25(OH)D. Adjusted for age, sex, BMI, hypertension, diabetes, smoking status, alcohol drinking status and blood collection month, SBP, DBP, TG, HDL-C, FBG, UA, Cr, HbA1c, and Hcy.

standard deviations above the mean for the same sex and age group [19–21]. The findings revealed a prevalence of EVA at 33.56%, which was significantly higher in males and associated with older average age in the EVA group. Compared with the control group, the EVA group had higher proportions of hypertension, diabetes, smokers, and alcohol drinkers, as well as higher levels of BMI, SBP, DBP, TG, TC, LDL-C, FBG, HbA1c, Cr, UA, and Hcy, which is consistent with previous studies [28–30].

Vitamin D is primarily synthesized by the skin upon exposure to sunlight, with dietary intake contributing minimally. The most prevalent circulating metabolite of vitamin D is 25-hydroxyvitamin D (25(OH)D), which is commonly regarded as the preferred indicator of serum vitamin D levels owing to its extended plasma half-life and robust stability [31]. Findings from the UK Biobank study indicated that a 4 ng/mL increase in serum 25(OH)D concentration was associated with a 12% decrease in all-cause mortality risk and a 9% decrease in CVD mortality risk [32]. Additionally, a meta-analysis confirmed

that low 25(OH)D levels could serve as an independent predictor of cardiovascular or all-cause mortality and major adverse cardiovascular events in patients with CVD [33]. Nevertheless, recent clinical intervention trials demonstrated that vitamin D supplementation did not significantly improve cardiovascular disease-related conditions [17,34]. This study aims to investigate the association between vitamin D and EVA, building upon recent findings that challenge the cardiovascular benefits of vitamin D. Findings indicated a significantly lower 25(OH)D level in the EVA group compared to the control group. Logistic regression analysis further suggesting a potential inverse association between 25(OH)D levels and EVA. Notably, quartile analysis demonstrated that individuals in the high-level group (Q4) exhibited a significantly reduced risk of EVA compared to those in the low-level group (Q1) (OR=0.71, P = 0.013), suggesting a potential non-linear relationship between the variables.

Subsequent analysis using RCS validated a significant non-linear relationship between 25(OH)D levels and EVA risk. Specifically, EVAsignificantly decreased with increasing 25(OH)D concentrations when levels were below 17.9 ng/mL. Beyond this threshold, this trend plateaued. Hence, it is imperative to acknowledge the critical role of serum 25(OH)D in the threshold effect of EVA. Several recent randomized controlled trials have indicated that vitamin D supplementation does not yield substantial additional health advantages for individuals with higher vitamin D levels (>20 ng/mL) [35]. Similarly, findings from Dai's significant prospective cohort study revealed a decline in the risk of all-cause mortality and CVD mortality as serum 25(OH)D concentrations increased; however, this trend plateaued at around 20 ng/mL [32]. The 25(OH)D threshold calculated in this study is 17.9 ng/mL, which is close to the aforementioned research results. This finding suggests that the association between lower serum 25(OH)D levels and EVA is more pronounced in populations with 25(OH)D concentrations < 17.9 ng/mL.

The inverse association between serum 25(OH)D and EVA can be explained through several mechanisms.. Firstly, it mitigates chronic inflammation in arterial walls by suppressing the NF-κB pathway and the synthesis of proinflammatory cytokines like IL-1β [36]. Secondly, it notably diminishes oxidative stress levels and augments nitric oxide production, thereby ameliorating vascular endothelial function [37]. Moreover, 25(OH)D suppresses the renin-angiotensin-aldosterone system's activity and facilitates the remodeling of vascular walls [38]. Recent research has also indicated that vitamin D curtails foam cell formation in VSMC and hinders the progression of atherosclerosis through the JNK-TLR4 signaling pathway [39]. Furthermore, vitamin D ameliorates insulin resistance, reduces serum total cholesterol and triglyceride levels, consequently enhancing glucose and lipid metabolism [40]. These mechanisms collectively establish a potential biological foundation for the prevention of EVA.

Our subgroup and interaction analyses revealed a significant inverse association between serum 25(OH)D levels and EVA in men after adjusting for confounding factors, whereas no significant association was observed in women, consistent with previous studies [41]. This sex-specific discrepancy may stem from biological differences, including sex hormone effects on lipid metabolism and vitamin D signaling pathways, as well as lifestyle variations between men and women. Additionally, the observed difference could be partly related to sample size distribution across subgroups, warranting further validation in larger, more balanced cohorts. Furthermore, a significant inverse association was observed between serum vitamin D and EVA in overweight and obese individuals, which is consistent with the findings of Zhang et al [38]. Mechanistically, serum 25(OH)D may effectively reduce obesity-related EVA by suppressing inflammatory factors, improving endothelial function, and enhancing insulin sensitivity. Interaction analysis indicated that, in addition to sex and BMI, there was also a statistically significant interaction in the age strata regarding the relationship between serum 25(OH)D levels and EVA.

Although the data collection process excluded the effects of diseases and medications on serum 25(OH)D levels and accounted for the blood collection month as a covariate to reduce the seasonal impact on 25(OH)D levels, other factors such as dietary intake and sun exposure were not considered, which may interfere with the research findings to some extent. Additionally, the exclusion of data related to diseases and medications affecting serum 25(OH)D levels was not subjected to sensitivity analysis, which may impact the robustness of the results. Although the predictive validity of baPWV

measurement for cardiovascular risk has been well established [4,18], the validity of the built-in reference curves as normal values for the Chinese population has not been specifically validated in independent studies, as detailed demographic characteristics of the source population for these reference curves are only available in device technical specifications and manuals, lacking direct support from scientific literature. However, the use of these reference curves remains reasonable given the consistency between our study cohort and the reference population in terms of ethnicity and selection criteria. This study is a cross-sectional design, and therefore, it cannot establish a causal relationship between serum 25(OH)D levels and the occurrence of EVA. Furthermore, the study population is derived from a single center, introducing potential selection bias and limiting the generalizability of the findings to other populations. Future research should involve multi-center, large-sample, prospective studies to further explore the intrinsic associations and mechanisms between these variables.

## Conclusion

The results of this study observed an inverse association between serum 25(OH)D levels and EVA, particularly with a more pronounced association possibly observed in men and individuals with overweight/obesity. Furthermore, we identified a threshold pattern for 25(OH)D, whereby levels below 17.9 ng/mL were associated with a significantly stronger inverse relationship with EVA. This study provides preliminary clues for the early identification of EVA and its potential relationship with vitamin D status, suggesting the necessity for further investigation of EVA-related factors in specific populations.

## Supporting information

**S1 Dataset. Raw data underlying the findings of this study.**
(XLSX)

**S2 File. Supplementary figures and table.**
(DOCX)

## Author contributions

**Conceptualization:** Jinpeng Cong, rui Hu.

**Data curation:** Jinpeng Cong, Jinyan Ren, Xinfeng Wang, Na Li.

**Formal analysis:** Jinpeng Cong, rui Hu.

**Methodology:** Jinpeng Cong.

**Software:** Jinpeng Cong, Ying Sun.

**Supervision:** rui Hu.

**Validation:** Jinyan Ren, Xinfeng Wang, Ying Sun.

**Visualization:** Na Li.

**Writing – original draft:** Jinpeng Cong.

**Writing – review & editing:** rui Hu.

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
