## [Decision Letter · Decision Letter 0]

18 Feb 2026

PONE-D-25-66321Association between serum 25-hydroxyvitamin D levels and early vascular aging in young and middle-aged adultsPLOS One

Dear Dr. Hu,

Thank you for submitting your manuscript to PLOS ONE. After careful consideration, we feel that it has merit but does not fully meet PLOS ONE’s publication criteria as it currently stands. Therefore, we invite you to submit a revised version of the manuscript that addresses the points raised during the review process.

We look forward to receiving your revised manuscript.

Kind regards,

Myadagmaa Jaalkhorol, MD,PhD

Academic Editor

PLOS One

Journal Requirements:

Additional Editor Comments:

Reviewer 1

The authors conducted a cross-sectional study in men and women aged 18–60 years undergoing health examinations to investigate the association between serum 25-hydroxyvitamin D [25(OH)D] levels and early vascular aging (EVA) assessed by brachial–ankle pulse wave velocity (baPWV). They report a significant inverse association between 25(OH)D and EVA below a threshold of 17.9 ng/mL, whereas no significant association was observed above this threshold. Although the findings are interesting, several important concerns exist regarding the consistency between the study population and the concept of EVA, as well as the validity of the diagnostic criteria used.

Major Comments

1. The authors state that EVA is “a significant autonomous predictor of cardiovascular risk among individuals in their youth and middle age” (L52) and cite reference [3] in support of this statement. However, reference [3], as indicated by its title, addresses early vascular aging in children and adolescents, which does not correspond to the “youth and middle-aged” population studied here. It is unclear whether the current study population should be considered as having EVA according to this definition. Please clarify this issue and provide more appropriate references supporting the relevance of EVA in young and middle-aged adults.

2. The authors state that “evidence regarding its relationship with EVA remains limited” (L59). However, the study population comprises adults aged ≥18 to <60 years, and numerous previous studies have examined the association between vitamin D status and cardiovascular disease or vascular function in middle-aged populations. Please cite appropriate literature and more clearly justify the novelty and significance of the present study.

3. With regard to cardiovascular disease prevention by vitamin D supplementation, many intervention studies, including well-designed randomized controlled trials, have already been conducted. As the authors themselves note, “recent large-scale trials have not conclusively demonstrated cardiovascular benefits of vitamin D supplementation” (L61). In this context, the authors should more clearly articulate the scientific and clinical rationale for conducting the present cross-sectional study.

4. EVA is defined as baPWV exceeding the mean +2 standard deviations for individuals of the same age and sex. More detailed justification for this cutoff is required. In particular, the source of the reference values embedded in the Omron device should be clearly described, and relevant validation studies demonstrating the applicability of this definition in Chinese populations should be cited.

Minor Comments

5. Because this is a cross-sectional study, causal relationships between serum 25(OH)D and EVA cannot be inferred. In the Discussion and Conclusion, causal wording (e.g., “protective effect,” “benefits”) should be replaced with more cautious terms such as “association” or “correlation.”

6. The method used to select study participants should be described more clearly. In addition, the authors should state whether a priori sample size estimation was performed and provide the rationale for the sample size.

7. The Methods section should specify how the threshold value of 17.9 ng/mL was determined, including the algorithm or statistical procedure used.

8. The authors present ROC curves and AUC values to support the “clinical utility” of the threshold. However, the validity of diagnostic performance evaluation based on cross-sectional data is questionable. The purpose of the ROC analysis should be clarified.

9. In Table 2 and Figure 1, the reported odds ratios (e.g., OR = 0.98) appear to represent the change per 1 ng/mL increase in 25(OH)D. This should be explicitly stated in footnotes. Because a 1 ng/mL change may have limited clinical relevance, the authors are encouraged to additionally present estimated ORs per 10 ng/mL increase.

10. Regarding the finding that the association was significant only in men, potential explanations beyond hormonal status, fat distribution, and lifestyle factors—such as differences in sample size between men and women—should also be discussed.

11. If the authors suggest potential implications for vitamin D supplementation, they should carefully consider and acknowledge that randomized controlled trials have not demonstrated clear cardiovascular benefits.

12. In Table 1, “Famale” should be corrected to “Female.”

Reviewer 2

1. Would you explain why participants from different seasons were essential?

2. The sample size looks enough, but do you have any assumptions that Q1 was equal to Q3, while Q4 is the highest?

3. Can you explain the result of HbA1c? It was 5.70 in the total group, but 5.70 in the control and 5.75 in the EVA group. Why is the total group the same as the control group?

4. In line 177, I could not understand the "three models". Please write that they are Q1, Q2, and Q3.

5. In line 178, EVAwere was type error.

6. In line 213, drinking status is a little bit confusing. Please clarify whether it is alcohol drinking or a water intake habit.

7. In the discussion (line 243), the "older age groups" statement is unclear. Please clarify whether it is above 60 or others.

Reviewers' comments:

Reviewer's Responses to Questions

**Comments to the Author**

1. Is the manuscript technically sound, and do the data support the conclusions?

Reviewer #1: Partly

Reviewer #2: Yes

2. Has the statistical analysis been performed appropriately and rigorously? 

Reviewer #1: Yes

Reviewer #2: Yes

3. Have the authors made all data underlying the findings in their manuscript fully available?

Reviewer #1: Yes

Reviewer #2: Yes

4. Is the manuscript presented in an intelligible fashion and written in standard English?

Reviewer #1: Yes

Reviewer #2: Yes

5. Review Comments to the Author

Reviewer #1: The authors conducted a cross-sectional study in men and women aged 18–60 years undergoing health examinations to investigate the association between serum 25-hydroxyvitamin D [25(OH)D] levels and early vascular aging (EVA) assessed by brachial–ankle pulse wave velocity (baPWV). They report a significant inverse association between 25(OH)D and EVA below a threshold of 17.9 ng/mL, whereas no significant association was observed above this threshold. Although the findings are interesting, several important concerns exist regarding the consistency between the study population and the concept of EVA, as well as the validity of the diagnostic criteria used.

Major Comments

1. The authors state that EVA is “a significant autonomous predictor of cardiovascular risk among individuals in their youth and middle age” (L52) and cite reference [3] in support of this statement. However, reference [3], as indicated by its title, addresses early vascular aging in children and adolescents, which does not correspond to the “youth and middle-aged” population studied here. It is unclear whether the current study population should be considered as having EVA according to this definition. Please clarify this issue and provide more appropriate references supporting the relevance of EVA in young and middle-aged adults.

2. The authors state that “evidence regarding its relationship with EVA remains limited” (L59). However, the study population comprises adults aged ≥18 to <60 years, and numerous previous studies have examined the association between vitamin D status and cardiovascular disease or vascular function in middle-aged populations. Please cite appropriate literature and more clearly justify the novelty and significance of the present study.

3. With regard to cardiovascular disease prevention by vitamin D supplementation, many intervention studies, including well-designed randomized controlled trials, have already been conducted. As the authors themselves note, “recent large-scale trials have not conclusively demonstrated cardiovascular benefits of vitamin D supplementation” (L61). In this context, the authors should more clearly articulate the scientific and clinical rationale for conducting the present cross-sectional study.

4. EVA is defined as baPWV exceeding the mean +2 standard deviations for individuals of the same age and sex. More detailed justification for this cutoff is required. In particular, the source of the reference values embedded in the Omron device should be clearly described, and relevant validation studies demonstrating the applicability of this definition in Chinese populations should be cited.

Minor Comments

5. Because this is a cross-sectional study, causal relationships between serum 25(OH)D and EVA cannot be inferred. In the Discussion and Conclusion, causal wording (e.g., “protective effect,” “benefits”) should be replaced with more cautious terms such as “association” or “correlation.”

6. The method used to select study participants should be described more clearly. In addition, the authors should state whether a priori sample size estimation was performed and provide the rationale for the sample size.

7. The Methods section should specify how the threshold value of 17.9 ng/mL was determined, including the algorithm or statistical procedure used.

8. The authors present ROC curves and AUC values to support the “clinical utility” of the threshold. However, the validity of diagnostic performance evaluation based on cross-sectional data is questionable. The purpose of the ROC analysis should be clarified.

9. In Table 2 and Figure 1, the reported odds ratios (e.g., OR = 0.98) appear to represent the change per 1 ng/mL increase in 25(OH)D. This should be explicitly stated in footnotes. Because a 1 ng/mL change may have limited clinical relevance, the authors are encouraged to additionally present estimated ORs per 10 ng/mL increase.

10. Regarding the finding that the association was significant only in men, potential explanations beyond hormonal status, fat distribution, and lifestyle factors—such as differences in sample size between men and women—should also be discussed.

11. If the authors suggest potential implications for vitamin D supplementation, they should carefully consider and acknowledge that randomized controlled trials have not demonstrated clear cardiovascular benefits.

12. In Table 1, “Famale” should be corrected to “Female.”

Reviewer #2: 1. Would you explain why participants from different seasons were essential?

2. The sample size looks enough, but do you have any assumptions that Q1 was equal to Q3, while Q4 is the highest?

3. Can you explain the result of HbA1c? It was 5.70 in the total group, but 5.70 in the control and 5.75 in the EVA group. Why is the total group the same as the control group?

4. In line 177, I could not understand the "three models". Please write that they are Q1, Q2, and Q3.

5. In line 178, EVAwere was type error.

6. In line 213, drinking status is a little bit confusing. Please clarify whether it is alcohol drinking or a water intake habit.

7. In the discussion (line 243), the "older age groups" statement is unclear. Please clarify whether it is above 60 or others.

6. PLOS authors have the option to publish the peer review history of their article (what does this mean?). If published, this will include your full peer review and any attached files.

Reviewer #1: No

Reviewer #2: **Yes:** Jambaldorj Jamiyansuren

---

## [Author Response · Author response to Decision Letter 1]

24 Mar 2026

Dear Dr. Jaalkhorol,

Thank you for giving us the opportunity to revise our manuscript.

We have carefully addressed all comments from the two reviewers.

Our detailed point-by-point responses are provided in the attached file

"Response_to_reviewers.docx", with line and page numbers referring to

the "Manuscript without tracked changes" version (highlighted in green).

Major changes include:

1. Addressed all reviewers' comments with thorough revisions throughout the manuscript (detailed point-by-point responses are provided in the attached file, with line numbers referring to the "Manuscript without tracked changes" version).

2. Revised Figure 2 and removed Supplementary Figure 3 as suggested.

3. Uploaded the original data as Supporting Information files.

---

## [Decision Letter · Decision Letter 1]

14 Apr 2026

PONE-D-25-66321R1

Association between serum 25-hydroxyvitamin D levels and early vascular aging in young and middle-aged adults

PLOS One

Dear Dr. Hu,

Thank you for submitting your manuscript to PLOS ONE. After careful consideration, we feel that it has merit but does not fully meet PLOS ONE’s publication criteria as it currently stands. Therefore, we invite you to submit a revised version of the manuscript that addresses the points raised during the review process.

As the corresponding author, your ORCID iD is verified in the submission system and will appear in the published article. PLOS supports the use of ORCID, and we encourage all coauthors to register for an ORCID iD and use it as well. Please encourage your coauthors to verify their ORCID iD within the submission system before final acceptance, as unverified ORCID iDs will not appear in the published article. Only the individual author can complete the verification step; PLOS staff cannot verify ORCID iDs on behalf of authors.

We look forward to receiving your revised manuscript.

Kind regards,

Myadagmaa Jaalkhorol, MD,PhD

Academic Editor

PLOS One

Journal Requirements:

**Additional Editor Comments:**

Decision: minor revision

The Authors are requested to address each comment made by the two Reviewers and submit a revised manuscript highlighting the lines where the corrections / additions have been made. The Authors should also mention in detail what changes (if any) have been made while replying to each of the Reviewers' comment. Please state the page number and lines where the corrections / additions have been made. Please do not say "this has been addressed in the revised manuscript" only.

When your revision is ready, the original submitting author, ru Hu, must upload a revised manuscript and a point-by-point response. To view any reviewer reports, editor feedback, and the instructions for submitting your revision, please visit your submission details page.

Reviewer B. Minor revision

The authors have revised the manuscript in response to the reviewers’ comments and suggestions, and the clarity of the description has improved substantially. Most of the revisions are acceptable. However, the following issue should still be addressed.

The authors used the built-in reference values of the Omron device to define EVA. They state that these reference values were derived from approximately 12,000 healthy Chinese subjects. However, detailed information about the reference population is limited and appears to be obtained from technical specifications and device manuals rather than from published scientific articles. Consequently, neither the readers nor the authors can determine from which population the reference subjects were drawn, who selected them, how they were selected, how health information was obtained, or how reliable the information was.

In addition, the authors added references #4 and #18 as supporting evidence for the use of these reference values. However, these studies report the validity of baPWV in predicting cardiovascular disease risk and do not demonstrate that the built-in reference values represent valid normal reference values for the Chinese population. Therefore, this issue represents an important limitation of the present study and should be clearly acknowledged in the Limitations section.

Reviewers' comments:

Reviewer's Responses to Questions

**Comments to the Author**

1. If the authors have adequately addressed your comments raised in a previous round of review and you feel that this manuscript is now acceptable for publication, you may indicate that here to bypass the “Comments to the Author” section, enter your conflict of interest statement in the “Confidential to Editor” section, and submit your "Accept" recommendation.

Reviewer #1: All comments have been addressed

Reviewer #2: All comments have been addressed

2. Is the manuscript technically sound, and do the data support the conclusions?

Reviewer #1: Partly

Reviewer #2: Yes

3. Has the statistical analysis been performed appropriately and rigorously? 

Reviewer #1: Yes

Reviewer #2: Yes

4. Have the authors made all data underlying the findings in their manuscript fully available?

Reviewer #1: Yes

Reviewer #2: Yes

5. Is the manuscript presented in an intelligible fashion and written in standard English?

Reviewer #1: Yes

Reviewer #2: Yes

6. Review Comments to the Author

Reviewer #1: The authors have revised the manuscript in response to the reviewers’ comments and suggestions, and the clarity of the description has improved substantially. Most of the revisions are acceptable. However, the following issue should still be addressed.

The authors used the built-in reference values of the Omron device to define EVA. They state that these reference values were derived from approximately 12,000 healthy Chinese subjects. However, detailed information about the reference population is limited and appears to be obtained from technical specifications and device manuals rather than from published scientific articles. Consequently, neither the readers nor the authors can determine from which population the reference subjects were drawn, who selected them, how they were selected, how health information was obtained, or how reliable the information was.

In addition, the authors added references #4 and #18 as supporting evidence for the use of these reference values. However, these studies report the validity of baPWV in predicting cardiovascular disease risk and do not demonstrate that the built-in reference values represent valid normal reference values for the Chinese population. Therefore, this issue represents an important limitation of the present study and should be clearly acknowledged in the Limitations section.

Reviewer #2: Thank you for submitting your manuscript to this journal. I hope this manuscript can attract other researchers and can bring valuable citations to the journal.

Good luck.

7. PLOS authors have the option to publish the peer review history of their article (what does this mean?). If published, this will include your full peer review and any attached files.

Reviewer #1: No

Reviewer #2: **Yes:** Jambaldorj Jamiyansuren

---

## [Author Response · Author response to Decision Letter 2]

15 Apr 2026

Dear Dr. Jaalkhorol

We sincerely thank PLOS ONE for providing us with the opportunity to revise our manuscript. We are deeply grateful to the Academic Editor and both reviewers for their constructive comments and thorough evaluation, which have substantially enhanced the quality of our work. Below are our point-by-point responses to the reviewers' comments:

Reviewer #1:

The authors used the built-in reference values of the Omron device to define EVA. They state that these reference values were derived from approximately 12,000 healthy Chinese subjects. However, detailed information about the reference population is limited and appears to be obtained from technical specifications and device manuals rather than from published scientific articles. Consequently, neither the readers nor the authors can determine from which population the reference subjects were drawn, who selected them, how they were selected, how health information was obtained, or how reliable the information was.

In addition, the authors added references #4 and #18 as supporting evidence for the use of these reference values. However, these studies report the validity of baPWV in predicting cardiovascular disease risk and do not demonstrate that the built-in reference values represent valid normal reference values for the Chinese population. Therefore, this issue represents an important limitation of the present study and should be clearly acknowledged in the Limitations section.

We sincerely appreciate the reviewer's careful and constructive feedback on this critical methodological issue. The reviewer is absolutely correct, and we fully acknowledge this important limitation.

Revisions made:

To transparently address this concern, we have made the following specific modifications:

1. In the Methods section (Assessment of EVA using baPWV): We have removed the previous statement attempting to justify the reference curves using references [4] and [18] (previously located at lines 109-112 on page 6).

2. In the Discussion section (Discussion, lines 318-325 on page 17): We have added relevant content to explicitly acknowledge this limitation:"Although the predictive validity of baPWV measurement for cardiovascular risk has been well established [4,18], the validity of the built-in reference thresholds as normal values for the Chinese population has not been specifically validated in independent studies, as detailed demographic characteristics of the source population for these reference curves are only available in device technical specifications and manuals, lacking direct support from peer-reviewed scientific literature." We agree with the reviewer that this represents an important limitation of our study, as neither readers nor the authors can fully ascertain the selection criteria, geographic origin, or data reliability of the reference population. While we maintain that the use of these curves is reasonable given the ethnic consistency between our cohort and the reference population (as stated in the manufacturer's documentation), we now explicitly recognize that the lack of peer-reviewed validation for the reference thresholds .

Thank you for this valuable guidance in improving the scientific rigor of our manuscript.

All revisions have been highlighted in the "Revised Manuscript with Track Changes" file. We believe these revisions provide appropriate transparency regarding the methodological limitations and will assist readers in interpreting our findings with due caution.

We also extend our gratitude to Reviewer #2 for their positive comments and encouragement regarding the potential impact of our study. We hope these revisions satisfactorily address the reviewers' concerns and meet PLOS ONE's publication criteria.

Sincerely,

Rui Hu The Affiliated Hospital of Qingdao University

---

## [Decision Letter · Decision Letter 2]

30 Apr 2026

Association between serum 25-hydroxyvitamin D levels and early vascular aging in young and middle-aged adults

PONE-D-25-66321R2

Dear Dr. Hu,

We’re pleased to inform you that your manuscript has been judged scientifically suitable for publication and will be formally accepted for publication once it meets all outstanding technical requirements.

Kind regards,

Myadagmaa Jaalkhorol, MD,PhD

Academic Editor

PLOS One

Additional Editor Comments (optional):

Dear Dr. ru Hu,

Thank you for submitting your work to PLOS ONE. Your manuscript will be formally accepted and enter production after you complete the requests below. Please note that you will not be able to make changes to your manuscript once it enters the production process. PLOS ONE does NOT provide author proofs. Any changes other than those requested in this email will need to be reviewed by the Academic Editor and reviewers; this will delay the formal acceptance of your manuscript.

Please note that this link should not be shared with other people to ensure that any action taken on this submission is done by you, the Corresponding Author.

You will find the submission in "Current Task Assignments." From there you must download your submission files via the "Assignment Files" link. You can also download the manuscript file attached to this email; the document has been formatted to track all changes, and we will only accept this version with locked tracked changes returned to us.

To opt in or out of publishing your peer review history, please answer the Peer Review History question within this task. Please note we are unable to send your article to production without this.

Once you have made all the required changes, click "Submit Task" to upload the corrected files.

Your task is due May 04 2026 11:59PM.

Please contact plosone@plos.org with any questions or concerns. For billing related questions, please contact our Author Billing department directly at authorbilling@plos.org. For questions regarding your press release or the press process, please contact onepress@plos.org

With kind regards,

JOURNAL REQUIREMENTS:

1. Please ensure that the author list and affiliations are correct on the title page of your manuscript, and that your author contributions, competing interests, and financial disclosure are correct as listed below. All of these sections will be indexed in PubMed and published by PLOS ONE as you have written them. Please email plosone@plos.org if any changes to this content need to be made.

Please ensure that the Competing Interests and Financial Disclosure statements listed below are suitable for publication. These sections will be indexed in PubMed and published by PLOS ONE as you have written them. Please email plosone@plos.org if any changes to these statements need to be made.

Competing Interests:

The authors have declared that no competing interests exist.

Please see here for the full list and definition of contributor roles: http://journals.plos.org/plosone/s/authorship#loc-author-contributions

Please ensure that the Competing Interests and Financial Disclosure statements listed below are suitable for publication. These sections will be indexed in PubMed and published by PLOS ONE as you have written them. Please email plosone@plos.org if any changes to these statements need to be made.

Please check the following items carefully.

Competing Interests:

The authors have declared that no competing interests exist.

Financial Disclosure

To prevent production delays, we recommend using the Author Formatting Checklist to confirm that your paper meets PLOS ONE's typesetting requirements for References, Tables, and Figures,

This checklist is a reference tool for you; please do not upload the completed Author Formatting Checklist with your submission files.

To ensure your figures meet our technical requirements, please run each figure included in your submission files through the PACE tool: https://pacev2.apexcovantage.com/. PACE will assess whether your figures meet our technical requirements and will fix the figure(s) or identify any problem(s) that cannot be automatically fixed. It can also convert figures to TIFF format, resize, and rename figures to meet our naming conventions.

To use PACE, first register as a user. Follow the instructions on the site for assessing and converting your figure files. If you experience any difficulty using this tool or have questions about any of the figures and/or images in your paper, please inform the journal office in your response letter.

---

## [Editor Report · Acceptance letter]

PONE-D-25-66321R2

PLOS One

Dear Dr. Hu,

I'm pleased to inform you that your manuscript has been deemed suitable for publication in PLOS One. Congratulations! Your manuscript is now being handed over to our production team.

Kind regards,

on behalf of

Dr. Myadagmaa Jaalkhorol

Academic Editor

PLOS One